# GRAPHTEXT: GRAPH REASONING IN TEXT SPACE

## ABSTRACT

Large Language Models (LLMs) have gained the ability to assimilate human knowledge and facilitate natural language interactions with both humans and other LLMs. However, despite their impressive achievements, LLMs have not made significant advancements in the realm of graph machine learning. This limitation arises because graphs encapsulate distinct relational data, making it challenging to transform them into natural language that LLMs understand. In this paper, we bridge this gap with a novel framework, GRAPHTEXT, that translates graphs to natural language. GRAPHTEXT derives a graph-syntax tree for each graph that encapsulates both the node attributes and inter-node relationships. Traversal of the tree yields a graph text sequence, which is then processed by an LLM to treat graph tasks as text generation tasks. Notably, GRAPHTEXT offers multiple advantages. It introduces *training-free graph reasoning*: even without training on graph data, GRAPHTEXT with ChatGPT can achieve on par with, or even surpassing, the performance of supervised-trained graph neural networks through in-context learning (ICL). Furthermore, GRAPHTEXT paves the way for *interactive graph reasoning*, allowing both humans and LLMs to communicate with the model seamlessly using natural language. These capabilities underscore the vast, yet-to-be-explored potential of LLMs in the domain of graph machine learning.

## 1 INTRODUCTION

Language stands as a cornerstone of human civilization, acting as the primary medium for knowledge encoding, reasoning, and communication. Large language models (LLMs), pre-trained on extensive text corpora, have showcased remarkable reasoning skills (Brown et al., 2020; Bubeck et al., 2023). These LLMs can communicate via natural language both internally (Wei et al., 2022) and externally with humans or other LLMs (Li et al., 2023), demonstrating exceptional skills such as multi-step reasoning (Yao et al., 2023a), decision-making (Yao et al., 2023b; Liang et al., 2023), tool use (Schick et al., 2023), and multi-agent collaboration (Park et al., 2023; Hong et al., 2023).

**Motivation.** Despite the remarkable success of LLMs in handling natural languages, their application to other data modalities presents unique challenges, primarily because these data often lack straightforward transformation into sequential text. These challenges are especially severe when dealing with graph-structured data, as different graphs define structure and features in distinct ways. Therefore, existing efforts within the graph machine learning field commonly require the training of specific graph neural networks (GNNs) tailored to individual graphs (Kipf & Welling, 2017; Velickovic et al., 2018; Xu et al., 2019). Often, models trained on one graph cannot generalize to the unseen structure and feature representations of other graphs. Moreover, the gap between graphs and human languages hinders the application of natural language reasoning to facilitate graph reasoning.

In light of these limitations, a question arises: *can we derive a language for graph in natural language?* In this paper, we give an affirmative answer by proposing to use *tree* as an intermediary, elegantly bridging structured data and one-dimensional sequential language. Essentially, *a tree exhibits a hierarchical structure, and traversing it yields a one-dimensional sequence.* On top of that, as shown in Figure 1 (c), we propose a novel framework GRAPHTEXT, which takes graph data to build a graph-syntax tree. Traversing it results in a graph prompt expressed in natural language, allowing an LLM to approach graph reasoning as a text-generation task.

**Main contributions.** First, GRAPHTEXT serves as a flexible and general framework for graph reasoning. It can incorporate common inductive bias of GNNs, such as feature propagation and feature

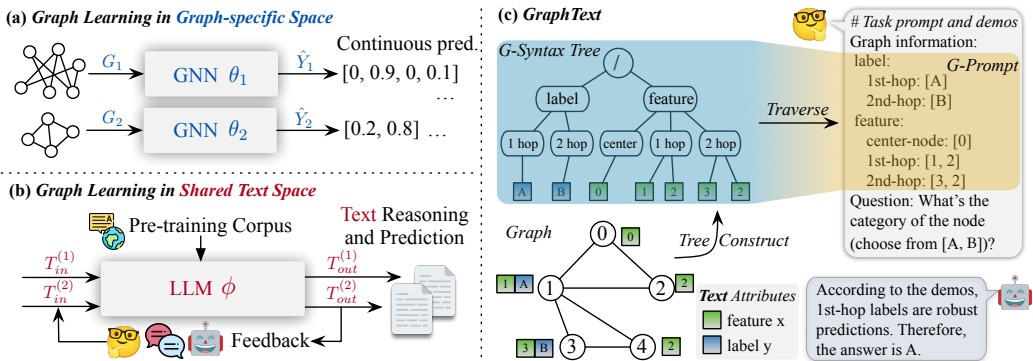

Figure 1: Comparison between **(a)** the GNN framework and **(b)** the proposed GRAPHTEXT framework. For different graphs $G_1$ and $G_2$, different GNNs $\theta_1$ $\theta_2$ are trained to make a graph-specific output prediction in continuous form. In contrast, GRAPHTEXT encodes the graph information to text sequences $T_{in}^{(1)}$ and $T_{in}^{(2)}$, and generates text reasoning and prediction $T_{out}^{(1)}$ and $T_{out}^{(2)}$ with a graph-shared LLM $\phi$. GRAPHTEXT leverages a pre-trained LLM to perform training-free graph reasoning and enables human and AI interaction for graph reasoning in natural language. **(c)** An example of the GRAPHTEXT framework that classifies node 0: Given a graph, GRAPHTEXT constructs a graph-syntax tree that contains both node attributes (e.g. feature and label) and relationships (e.g. center-node, 1st-hop, and 2nd-hop). Then, GRAPHTEXT traverses the graph-syntax tree to obtain a sequential text, i.e. graph prompt, and let LLM perform graph reasoning in text space.

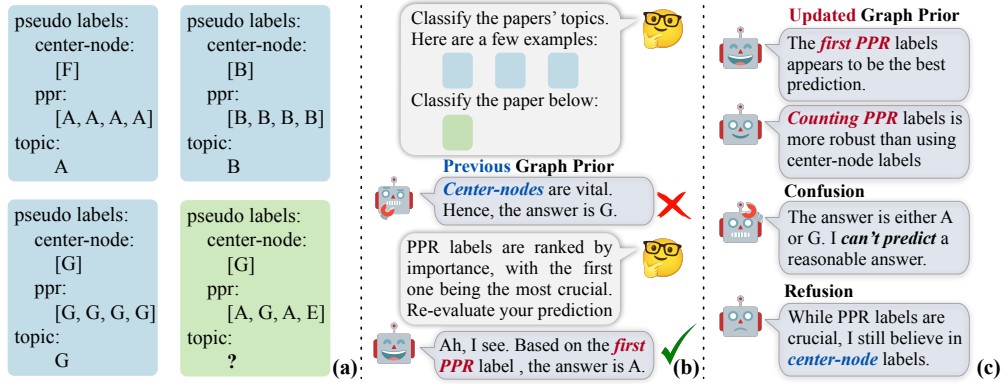

Figure 2: **(a)** Few-shot demonstrations (blue) and the target node #2188 (green) to predict on Cora. **(b)** An illustration of how human interaction changes the graph reasoning of an LLM, where the LLM previously has the prior that the center-node is vital. **(c)** Behaviors of LLMs after given demonstrations/human interaction: *update graph prior* to bias more on PPR (personalized pagerank); leads to *confusion* or *refusion*. Details are discussed in Section 4.2.

similarity-based propagation, simply by constructing different graph-syntax trees. It also serves as a general framework for graph reasoning for both in-context learning and instruction tuning, on both general graphs and text-attributed graphs. Second, we show that GRAPHTEXT enables the possibility of **training-free graph reasoning**. The training-free property enables us to deploy GRAPHTEXT not only with open-source LLMs, but also with powerful closed-source LLMs. Remarkably, even without training on graph data, GRAPHTEXT with ChatGPT can deliver performance on par with, or even surpassing, supervised graph neural networks through in-context learning. This highlights the vast potential of foundation models in the realm of graph machine learning. Third, GRAPHTEXT fosters **interactive graph reasoning**: With its capacity to generate and **explain** predictions in natural language, humans can directly engage with GRAPHTEXT. As shown in Figure 2 (b), through interactions with humans and other LLMs, GRAPHTEXT refines its graph reasoning capabilities.

## 2 METHODOLOGY

In this section, we introduce GRAPHTEXT to perform graph reasoning in text space. Out of the three fundamental problems of graph ML (graph classification, node classification, and link prediction), we take node classification as an example to introduce our idea. We however note that our discussion applies to other graph tasks.

### 2.1 THE GRAPHTEXT FRAMEWORK

Let us be given an attributed graph $G = (V, E, \boldsymbol{X})$ with nodes $V$ and edges $E$, whose structure is represented as the $|V| \times |V|$ adjacency matrix $\boldsymbol{A}$ and node attributes as the $|V| \times d$ feature matrix $\boldsymbol{X}$. Given a subset $L \subset V$ of labeled nodes with labels $Y_L$, the goal of node classification is to predict the labels $Y_U$ of the unlabeled nodes $U = V \setminus L$. Graph Neural Networks (GNNs) are the standard architecture for such problems. As shown in Figure 1 (a), a GNN directly learns a parametric map

$$\hat{y}_i = f_{\text{GNN}}(G; \theta_G)_i \tag{1}$$

between the input graph $G \in \mathcal{G}$ and the output labels $\hat{Y} \in \mathcal{Y}$, assigning to each node $i$ its predicted label $\hat{y}_i$. The training of GNN attempts to find parameters $\theta_G$ such that $\hat{y}_i \approx y_i$ on the training set. Note that standard GNNs are ***graph-specific*** functions, i.e. $f_{\text{GNN}}(\cdot; \theta_G) : G \mapsto \hat{Y}$, which do not generalize to other graphs, since other graphs $G' \in \mathcal{G}$ define distinct distributions of $Y'$, $\boldsymbol{A}'$, and $\boldsymbol{X}'$, or even different types of features such as continuous, categorical, or text features.

To solve the generalization problem mentioned above, this paper proposes to perform graph reasoning as a *text-to-text* problem (Raffel et al., 2020), as shown in Figure 1 (b). Inspired by prompt tuning (Brown et al., 2020; Liu et al., 2023), we construct two graph-specific maps to form the input and output space of a text-to-text problem: a map $g : G \mapsto T_{in}$ that maps the graph input to text space, and a map $h : T_{out} \mapsto \tilde{Y}$ that maps the output of LLM to label predictions $\tilde{Y}$. In this way, we can use a generative large language model $f_{\text{LLM}}$ to perform graph reasoning as

$$\tilde{y}_i = h(f_{\text{LLM}}(g(G)_i; \phi)) \tag{2}$$

where $g(G)_i = T_{in}[i]$ denotes the text sequence representing node $i$. Different from GNNs, $f_{\text{LLM}}(\cdot; \phi) : \mathcal{T} \to \mathcal{T}$ is a ***graph-shared*** function, where both input and output are in text space, i.e. $T_{in}, T_{out} \in \mathcal{T}$, which not only activates of parametric knowledge encoded in the model $f_{\text{LLM}}(\cdot; \phi)$, but also enables interactions between human and AI agents to facilitate graph reasoning.

Specifically, as node classification, link prediction and graph classification are essentially classification tasks, we can naturally formulate these graph reasoning tasks as multi-choice QA problem (Robinson & Wingate, 2023) and design $h$ as the map from predicted choice $T_{out} \in \mathcal{T}$ to the corresponding prediction $\tilde{Y}$. However, the design of $g$ that maps the structural graph information into the text space of natural language is still a non-trivial problem.

The primary challenge in converting graph data to language lies in handling its relational structure, which fundamentally deviates from the one-dimensional sequential nature of text data. Inspired by linguistic syntax trees (Chiswell & Hodges, 2007), we introduce graph-syntax trees as a bridge between relational and sequential data. The traversal of such a tree produces a sentence in natural language, which is fed to LLM for graph reasoning. Specifically, as shown in Figure 1 (c), we compose a graph-syntax tree consisting of node text attributes and inter-node relationships. Next, we describe how to compose the node text attributes and inter-node relationships in Section 2.2, and how to build a graph-syntax tree in Section 2.3.

### 2.2 TEXTUAL AND RELATIONAL INFORMATION FOR SYNTAX TREES

A graph syntax tree is composed of both textual and relational information derived from the graph. For textual information, GRAPHTEXT constructs a text attribute set $F \in \mathcal{T}$ for an arbitrary graph $G \in \mathcal{G}$ (with or without text-attributes) composed of multiple types of attributes for each node, e.g. feature and label, in natural language. Specifically, for each node $v_i$ and feature type $m$, we construct a text sequence $\mathrm{F}_m[i]$ in natural language:

$$\mathrm{F}_m[i] = \{t_1, t_2, \cdots t_{l_m}\}, \ \ \mathrm{F}_m[i] \in \mathcal{T}, \tag{3}$$

where the sequence is of length $l_m$. Each text attribute $F_m$ can be derived from either sequential text features or continuous features. For text features, they can be directly added to the text attributes $F$. For example, we can directly add the text sequences of "title" and "abstract" into $F$ for citation graphs. For continuous features, e.g. the raw feature $X$ or other graph embeddings, we propose to use discretization methods, e.g. clustering, to transform the continuous feature into a discrete space and then derive sequential data from it. For simplicity, we use the cluster index of K-means to generate a sequence of length 1 for all continuous features as K-means is effective in our experiments.

For relational information, GRAPHTEXT derives a set of matrices $R$ where each $R_n \in R$ is a $|V| \times |V|$ matrix, depicting one type of relationship between nodes. Choices of $R_n$ may be the original graph (Kipf & Welling, 2017), high-order connectedness (Xu et al., 2018), page-rank matrices (Klicpera et al., 2019), or any matrices that encode node-pair information. These relationships play an important role in determining the nodes and structure of the graph-syntax tree, which further improves the graph text prompt.

## 2.3 GRAPH-SYNTAX TREE COMPOSITION

We now describe how to build a graph prompt using a graph-syntax tree of the graph text attributes and relationships $F$ and $R$. By analogy to the syntax tree in linguistics, we define a graph-syntax tree as an ordered tree: a directed acyclic graph (DAG) with nodes $\tilde{T} \in \mathcal{T}$ and edges $\tilde{E}$. In a graph-syntax tree, e.g. the one in Figure 1 (c), each node stores a *text sequence* in natural language, where the root node is an empty node; the leaf nodes $\tilde{T}_L$ are text sequences in the graph text attributes, i.e. $\forall T_i \in \tilde{T}_L, T_i \in F$; the internal nodes $\tilde{T}_I$ are text sequences in natural language, i.e. $\forall T_i \in \tilde{T}_I, T_i \in \mathcal{T}$. A graph syntax tree is constructed in three steps: (1) construct an ego-subgraph (Hamilton et al., 2017) $G_i$ for target node $v_i$ based on relationship $R$ (2) select leaf nodes $\tilde{T}_L$ based on the relationship $R$. (3) build up internal nodes $\tilde{T}_I$ and edges $\tilde{E}$ based on the leaf nodes' types and their relationship with the graph[1]. Notably, the leaf nodes are sorted according to their relationships with the center-node, preserving the relative relationship in a one-dimensional order.

We illustrate this with a node classification example shown in Figure 1 (c). Before building the graph-syntax tree, GRAPHTEXT determines the text attributes composed of raw features and observed labels, i.e. $F = \{F_X[i], F_Y[i] \mid \forall v_i \in V\}$, and a relationship set composed of determined by shortest path distance (SPD): center-node, 1st-hop, and 2nd-hop, i.e. $R = \{R_{SPD=0}, R_{SPD=1}, R_{SPD=2}\}$. Then, for target node $v_i$ (0 in the example), an ego-subgraph (Hamilton et al., 2017) (with nodes [0,1,2,3,4]) is sampled based on the relative relationship between $v_i$ and other nodes. Finally, a graph-syntax tree is constructed with leaf nodes $\tilde{T}_L = \{F_X[0], F_X[1], F_X[2]F_X[3], F_X[4], F_Y[1], F_Y[3]\}$, the internal nodes $\tilde{T}_I = \{$"center-node", "1st-hop", "2nd-hop", "label", "feature"$\}$, and the corresponding edges. The traversal of the resulting graph-syntax tree leads to a text sequence in natural language.

Compared with the direct flattening of (sub)graphs (Wang et al., 2023a; Chen et al., 2023), using a graph-syntax tree-based prompt has many advantages: Above all, unlike a graph, which has no topology order, a syntax tree is a DAG that can be topologically sorted, which gracefully converts a relational structure to a sequence of nodes. Moreover, GRAPHTEXT easily incorporates the inductive biases of GNNs through the construction of node text attributes $F$ and relationships $R$. For example, we can easily encode the feature-propagation mechanism of GNNs by including a text attribute derived from the propagated feature $A^k X$ (Zhang et al., 2022), into the node attributes $F$. We can also incorporate the feature similarity-based aggregation (Velickovic et al., 2018) by adding $X X^\top$ to $R$. These graph-based inductive biases can significantly boost LLMs' graph reasoning performance (further discussed in Section 4.1). Last but not least, a tree naturally defines a hierarchical structure, which LLMs are proficient in reasoning on (Liang et al., 2023), by training on code data (Chen et al., 2021) and web page data (Touvron et al., 2023).

---

[1]The hierarchy of the tree can be defined flexibly, but we have empirically discovered that a simple configuration, with attribute type at the top hierarchy and relation type at the bottom hierarchy for internal nodes, as illustrated in Figure 1 (c), yields strong performance. Further details are available in Section 4.3.

## 3 RELATED WORK

**Unlock Graph Space for Language Models.** Large Language Models (LLMs) (Brown et al., 2020; OpenAI, 2023; Anil et al., 2023; Bubeck et al., 2023) possess impressive reasoning capabilities (Wei et al., 2022; Yao et al., 2023a; Fu et al., 2023). At the heart of LLMs' reasoning prowess is their ability to process and generate natural language inputs and outputs, enabling flexible interactions (Dohan et al., 2022) with both humans and AI agents. This unique capability empowers them with remarkable abilities such as complex reasoning (Fu et al., 2023) and decision-making (Yao et al., 2023b; Liang et al., 2023). Despite their success, applying LLMs to relational graph data remains challenging, primarily due to the absence of a natural language representation for graphs. GRAPHTEXT bridges this gap by providing a novel framework that enables LLMs to seamlessly integrate and reason over relational graph data using the same natural language capabilities, thereby unlocking their potential for a wide range of graph-based applications.

**Training-free Graph Reasoning** Graph neural networks (GNNs) (Kipf & Welling, 2017; Xu et al., 2019) excel in handling relational graph data, thanks to the message-passing mechanism for aggregation and transformation of neighborhood representations. Their standout performance can be attributed to their intrinsic capability to assimilate graph inductive biases. This incorporation of biases is achieved by designing representations with the graph structure in perspective, such as position embeddings (Dwivedi et al., 2022; Ying et al., 2021; Kreuzer et al., 2021) and propagated features (Wu et al., 2019; Zhang et al., 2022). Furthermore, they can introduce diverse aggregation methods, like feature similarity-based message passing (Velickovic et al., 2018; Zhao et al., 2021) or high-order aggregation (Klicpera et al., 2019; Bojchevski et al., 2020; Chien et al., 2021). However, as highlighted in Section 2.1, due to the variance in both structure and feature, the majority of GNNs are *graph-specific*. They are tailored for a particular graph type with consistent features and structures, thus posing challenges for generalization to different graphs.

In a parallel vein, GRAPHTEXT also taps into the potent ability to infuse graph inductive biases for graph reasoning, achieved through designing both the textual and relational aspects of the graph-syntax tree. Setting itself apart from GNNs, GRAPHTEXT approaches graph reasoning in a *graph-shared* domain, facilitating the broader applicability of a single LLM to diverse graphs and offering training-free and interactive graph reasoning.

**Connecting Both Worlds.** Recent endeavors (Chien et al., 2022; Zhao et al., 2023; He et al., 2023) have aimed to merge the language and graph domains. Most methods involve transitioning the problem into a graph-specific realm, utilizing a combination of a text-encoder (either pre-trained (Chien et al., 2022) or learned (Li et al., 2021)) and a GNN predictor. This methodology still falls into a *graph-specific* paradigm. Very recently, there are concurrent works (Guo et al., 2023; Ye et al., 2023; Wang et al., 2023a; Chen et al., 2023) exploring to leverage LLMs for graph-related tasks. These methods either directly flatten the nodes and edges (Guo et al., 2023; Wang et al., 2023a) or employ rule-based prompts on text-attributed graphs (Chen et al., 2023; Ye et al., 2023).

Nevertheless, GRAPHTEXT is fundamentally different from these works. Foremost, GRAPHTEXT proposes a language defined by a graph-syntax tree, offering a flexible and structured approach for seamlessly integrating graph inductive biases. Moreover, it also serves as a general framework for graph reasoning, which can be applied to scenarios encompassing in-context learning and instruction tuning. It accommodates various types of graphs, including general graphs and text-attributed graphs, and is adaptable to both closed-source Large Language Models (LLMs) (OpenAI, 2023; Bubeck et al., 2023) and open-source LLMs (Touvron et al., 2023).

## 4 EXPERIMENTS

We conduct extensive experiments to demonstrate the effectiveness of GRAPHTEXT. Firstly, in Section 4.1, we delve into the remarkable capacity of GRAPHTEXT for training-free graph reasoning. Subsequently, Section 4.2 highlights the interactive graph reasoning capabilities of GRAPHTEXT. We further analyze various ablations of graph-syntax trees in Section 4.3. Concluding our exploration, Section 4.4 illustrates how GRAPHTEXT can seamlessly function as a versatile framework, catering to both in-context learning and instruction tuning across on both general graph and text-attributed graphs.

Table 1: Node classification results (accuracy %). $\Delta$ GCN and $\Delta$ Best ICL denotes the performance gain of GRAPHTEXT over GCN and ICL baselines, respectively.

| Setting | Model | Cora | Citeseer | Texas | Wisconsin | Cornell |
|---------|-------|------|----------|-------|-----------|---------|
| Supervised Learning | GCN | 81.4 | 69.8 | 59.5 | 49.0 | 37.8 |
| | GAT | 80.8 | 69.4 | 54.1 | 49.0 | **45.9** |
| | GCNII | 81.2 | 69.8 | 56.8 | 51.0 | 40.5 |
| | GATv2 | **82.3** | **69.9** | **62.2** | **52.9** | 43.2 |
| In-Context Learning | NeighborText | 26.3 | 13.7 | 5.4 | 9.8 | 21.6 |
| | GML | 38.5 | 28.4 | 10.8 | 23.5 | 21.6 |
| | GraphML | 49.9 | 28.9 | 16.2 | 33.3 | 29.7 |
| | GRAPHTEXT ot+or | 33.4 | 36.9 | 5.4 | 29.4 | 24.3 |
| | GRAPHTEXT ot+sr | 52.1 | 50.4 | 73.0 | 60.8 | 46.0 |
| | GRAPHTEXT st+or | 64.5 | 51.0 | 73.0 | 35.3 | 48.7 |
| | GRAPHTEXT st+sr | **68.3** | **58.6** | **75.7** | **67.6** | **57.9** |
| Comparisons | $\Delta$ GCN | -13.1% | -11.2% | +16.2% | +18.6% | +20.1% |
| | $\Delta$ Best ICL | +18.4% | +29.7% | +59.5% | +34.3% | +28.2% |

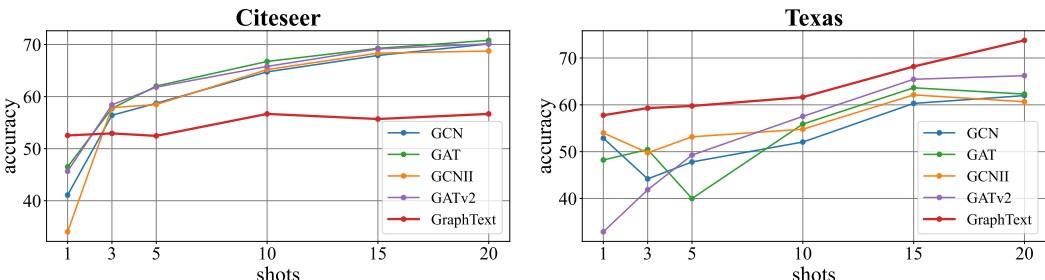

Figure 3: Few-shot in-context learning node classification accuracy. We perform 1, 3, 5, 10, 15, and 20-shot node classification on Citeseer and Texas datasets.

## 4.1 TRAINING-FREE GRAPH REASONING

One unique ability of GRAPHTEXT is the training-free graph reasoning by in-context learning (ICL). In this section, we demonstrate this capability on the node classification tasks. Specifically, we use two citation datasets (*Cora* (McCallum et al., 2000) and *Citeseer* (Giles et al., 1998)), and three webpage datasets (*Texas*, *Wisconsin*, and *Cornell* (Pei et al., 2020)). The detailed discussion of experimental settings and the dataset statistics can be found in Appendices A.1 and A.2.

We compare GRAPHTEXT with two types of baselines: (1) Supervised learning GNNs: including *GCN* (Kipf & Welling, 2017) and *GAT* (Velickovic et al., 2018), along with their more recent variants *GCNII* (Chen et al., 2020) and *GATv2* (Brody et al., 2022); (2) In-Context Learning (ICL) LLM baselines: NeighborText Chen et al. (2023) (the only available raw text attributes in the graph is the observed node labels), GML, and GraphML (Guo et al., 2023). The GNN baselines are supervised trained solely for inference on one dataset. In contrast, ICL baselines utilize a single pre-trained LLM (ChatGPT) for all datasets without any graph-specific training.

For GRAPHTEXT, we derive four ablations based on the text and relation information used when constructing the graph-syntax tree. For *text information*, the original graph information (feature, label) and synthetic text information (propagated feature/labels) are denoted as *ot* and *st*. For *relation information*, the original graph relation and the synthetic relations (e.g. shortest-path based, feature similarity-based, and personalized PageRank relations), denoted as *or* and *sr*, respectively.

The experimental findings, as shown in Table 1 and Figure 3, reveal that methods using the raw graph as a prompt, such as NeighborText, GML, GraphML, and GRAPHTEXT ot+or, exhibit subpar performance. However, the incorporation of graph inductive bias into both synthetic text attributes and relations leads to significant performance improvements across all datasets. Notably, GRAPHTEXT st+sr, integrating these elements, sets a new benchmark in graph in-context learning, outperforming

Table 2: Interactive graph reasoning results (accuracy %) on Cora (node # 2188). The table show-cases the performance of GPT-4 and ChatGPT before and after human interactions with 15 times of evaluation. The reasoning metrics include PPR, Center-node, and instances where the model was Confused to respond or Refused (Conf./Ref.) to make their reasoning/prediction. See Figure 2 (c) for details.

| Model | Interaction | Accuracy | Reasoning | | |
|---|---|---|---|---|---|
| | | | PPR | Center-node | Conf./Ref. |
| GPT-4 | Before | 73.3 | 73.3 | 26.7 | 0 |
| | After | **100 (+26.7)** | 100 | 0 | 0 |
| ChatGPT | Before | 26.7 | 26.7 | 53.3 | 20.0 |
| | After | **63.6 (+36.9)** | 72.7 | 18.2 | 9.1 |

the best ICL baseline by an average of 34.0%. Impressively, GRAPHTEXT occasionally **exceeds supervised learning GNN baselines** through in-context learning, especially in scenarios of low la-bel rates (as indicated in Figure 3) and within heterophilic datasets. This superior performance is attributed to GRAPHTEXT's ability to uncouple depth and scope in graph reasoning (Zeng et al., 2021), unlike traditional GNNs. The remarkable efficacy of GRAPHTEXT in training-free graph reasoning underscores the immense potential of LLMs in the realm of graph machine learning.

## 4.2 INTERPRETABLE AND INTERACTIVE GRAPH REASONING

In this section, we illustrate that GRAPHTEXT facilitates effective **interactive graph reasoning**: through its ability to generate and **clarify** predictions in natural language, both humans and LLMs can directly interact with GRAPHTEXT.

To illustrate this concept, we will use Cora node #2188. Figure 2 (a) shows two types of text at-tributes we use: the center-node pseudo labels and the PPR (Personalized PageRank) pseudo label sequence, where the first PPR neighbor denotes the most important label prediction. Upon examin-ing the demonstrations (marked in blue), it becomes apparent that the PPR pseudo-labels provide a more robust mechanism for paper topic prediction. Utilizing either a count of PPR labels followed by a majority vote, or merely referencing the foremost PPR label, consistently results in the correct categorization in the given examples. Hence, based on these graph inductive biases derived from samples, we can reasonably figure out correct topic of the target paper should be A, which not only is the first entry, but also the predominant label in the PPR pseudo-label sequence.

We leverage GRAPHTEXT with ChatGPT and GPT-4 to perform graph reasoning on the provided example. Their respective reasoning processes and outcomes are illustrated and summarized in Figure 2 and Table 4.1 respectively, from which we draw several key insights:

**1. LLMs inherently possess the knowledge and inductive bias toward graph reasoning.** Specif-ically, both ChatGPT and GPT-4 acknowledge the importance of center-nodes and sometimes make predictions based on center-node labels. ChatGPT exhibits reasoning with the center-node bias 53.3% of the time, while GPT-4 does so at a rate of 26.7%.

**2. LLMs can adjust their prior inductive bias based on demonstrations.** Through in-context learning, GRAPHTEXT can recalibrate their bias and make more accurate predictions. Our obser-vations indicate that GPT-4 significantly outperforms ChatGPT, achieving an accuracy of 73.3%, markedly superior to ChatGPT's 26.7%.

**3. LLMs can adapt their prior inductive bias based on human feedback.** Figure 2 (b) provides an illustrative example, with a detailed reasoning of LLM can be found in Appendix C. Specifically, after human interaction, GPT-4 shows remarkable adaptability, achieving an impeccable accuracy of 100% and adhering to the PPR logic. Meanwhile, ChatGPT also enhances its performance notably (gaining 36.9% in accuracy), but occasionally maintains its antecedent biases.

In summary, through graph reasoning in natural language, GRAPHTEXT can effectively leverage its pre-trained knowledge to engage in graph reasoning and, crucially, adapt its existing knowledge through demonstrations or external feedback.

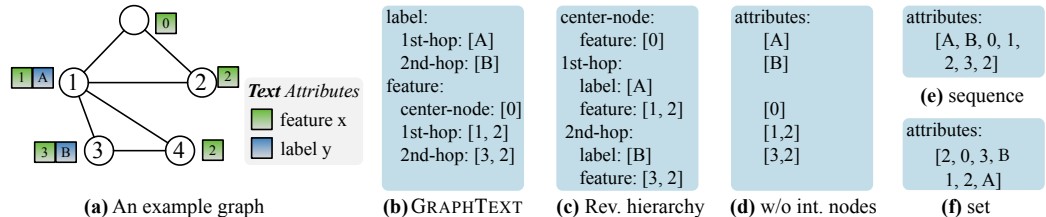

Figure 4: Ablations of graph-syntax trees. **(a)** An example graph. **(b)** GRAPHTEXT text prompt (The full example can be found in Figure 1). **(c-f)** Text prompts of different tree designs.

Table 3: Ablations of GRAPHTEXT on Cora, Citeseer and Texas.

| Model | Cora | | Citeseer | | Texas | |
|---|---|---|---|---|---|---|
| | Acc. % | Δ | Acc. % | Δ | Acc. % | Δ |
| **GRAPHTEXT** | **68.3** | - | **58.6** | - | **75.7** | - |
| rev. hierarchy | 68.3 | -0 % | 57.6 | -1.7 % | 73.0 | -3.6 % |
| w/o int. nodes | 67.8 | -0.5 % | 56.3 | -3.9 % | 75.7 | -0 % |
| sequence | 67.0 | -1.3 % | 53.0 | -9.6 % | 70.3 | -7.1 % |
| set | 65.9 | -2.4 % | 56.4 | -3.8 % | 67.6 | -10.6 % |

## 4.3 ABLATION STUDIES ON GRAPH-SYNTAX TREES

The graph-syntax tree serves as the core design of GRAPHTEXT, transforming a graph into a 1-dimensional natural language sequence. Within GRAPHTEXT, the text and relational data of a graph are initially formulated, followed by the construction of a graph-syntax tree. This section delves into the ablations of various methods for building graph-syntax trees.

As shown in Figure 4, besides the proposed GRAPHTEXT method for constructing a graph-syntax tree, we present four ablation types: (1) Reverse hierarchy (denoted as **rev. hierarchy** in Figure 4 (c): The tree hierarchy is inverted, positioning the relationship type at the top and the text attribute type at the bottom. (2) Without internal nodes (denoted as **w/o int. nodes** in Figure 4 (d)): The internal nodes of the graph-syntax tree are eliminated, but the GRAPHTEXT hierarchy remains intact (note the indents are kept, maintaining the hierarchical structure of the tree). (3) Sequential prompt (denoted as **sequence** in Figure 4 (e)): The tree hierarchy is removed, yielding a sequence of text attributes. (4) Set prompt (denoted as **set** in Figure 4 (f)): Sequence order is removed, yielding a set.

From Table 3, several observations can be made: (1) The graph-syntax tree of GRAPHTEXT consistently outperforms the others, underscoring the efficacy of our approach. (2) The hierarchical structure of the tree plays a crucial role in the design of the graph prompt. Specifically, we observe a sheer performance drop when using a *sequence* or a *set* to represent the graph information. Upon inspecting the LLM's reasoning, we found it treats graph learning purely as label counting without recognizing structure. (3) Variations in the tree hierarchy design can impact performance; for instance, *rev. hierarchy* underperforms compared to GRAPHTEXT. (4) The comparison between *w/o int. nodes* and GRAPHTEXT reveals the importance of making LLMs aware of text attribute types. The only exception is the Texas dataset since all types of attributes are almost identical in this dataset (detailedly discussed in Appendix B.3). This suggests that LLMs utilize text descriptions to distinguish and understand different attributes during graph reasoning.

## 4.4 EXPERIMENTS ON TEXT-ATTRIBUTED GRAPH

In this section, we demonstrate that GRAPHTEXT is also applicable to text-attributed graphs. As depicted in Table 4, we conducted training-free node classification on the Cora and Citeseer datasets with both raw text attributes (Chen et al., 2023) and continuous features (Reimers & Gurevych, 2019). We observed that using closed-source LLMs, such as ChatGPT, the performance lags behind the GNN baseline methods. Thus, we further explored the potential of instruction tuning on currently available open-source LLMs, such as Llama-2 (Touvron et al., 2023). For natural language prompts

Table 4: Node classification results (accuracy %) on real-world text attributed graphs. Experiments are conducted using in-context learning with ChatGPT, as well as instruction tuning with Llama-2-7B. Note that "text" refers to raw text attributes, while "feat" represents the continuous features on the graph. The top results for each category are highlighted in bold.

| Framework | Model | Cora | Citeseer |
|---|---|---|---|
| GNNs | GCN | 89.13 | 74.92 |
| | GAT | **89.68** | **75.39** |
| GRAPHTEXT | ChatGPT-text | **67.77** | **68.98** |
| | ChatGPT-feat | 10.68 | 16.14 |
| | ChatGPT-feat+text | 65.19 | 66.46 |
| | Llama-2-7B-text | 60.59 | 49.37 |
| | Llama-2-7B-feat | **87.11** | **74.77** |
| | Llama-2-7B-feat+text | 77.53 | 73.83 |

construction, we adopted an approach almost identical to the in-context learning setting. Furthermore, we expand the original vocabulary of Llama-2 by introducing selected options as new tokens and then fine-tune the large language model by the widely-used and efficient Low-Rank Adaptation (LoRA) (Hu et al., 2022).

From the results in Table 4, it is evident that even with a relatively smaller open-source model, Llama-2-7B, our best results from instruction tuning across various settings surpass those of Chat-GPT and approach the GNN baselines. This validates that our method can be beneficial in an instruction-tuning scenario. It also implies that using GRAPHTEXT, we can feasibly fine-tune smaller open-source LLMs with reasonable computational costs, achieving performances that can rival or even surpass those of much larger closed-source models, such as ChatGPT or GPT-4.

Another intriguing observation is the notably poor performance of ChatGPT in settings incorporating continuous feature – nearing a random guess. This is attributable to the inherent limitation of these closed-source LLMs: they are designed to process raw discrete text inputs and fail to directly handle the continuous inputs. In contrast, open-source LLMs possess the ability to map these continuous embeddings into their embedding space, facilitating improved performance.

Upon contrasting these two groups of models, we noticed a decline in the performance of open-source models when processing raw text inputs. This decline can be ascribed to the constraints imposed by the size of the LLM parameters and the volume of pre-training corpora used. It suggests that harnessing larger-scale open-source models, such as Llama-2 variants including 13B, 30B, and 70B, would significantly bolster their modeling capacity for raw text. Concurrently, by leveraging the ability to process continuous embeddings, these models would inevitably exhibit enhanced graph reasoning capabilities, paving the way for more sophisticated graph-based applications.

## 5 CONCLUSION

In this paper, we propose GRAPHTEXT, a framework that enables graph reasoning in text space. It easily incorporates the inductive bias of GNNs by constructing a graph-syntax tree. The traversal of a graph-syntax tree leads to a graph prompt in natural language and is fed to LLM to perform graph reasoning as text generation. GRAPHTEXT enables training-free graph reasoning where a GRAPHTEXT-LLM can deliver performance on par with, or even surpassing, supervised graph neural networks through in-context learning. What's more, GRAPHTEXT fosters explainable and interactive graph reasoning: GRAPHTEXT performs graph reasoning in natural language which enables humans and LLMs to engage with graph learning using natural language. These abilities highlight the immense and largely untapped potential of LLMs in the realm of graph machine learning.

ETHICS STATEMENT

Graphs are prevalent in the real world. On the bright side, GRAPHTEXT alleviates the computational load and carbon footprint associated with training numerous non-transferable, graph-specific models. However, while the training-free graph reasoning capability of GRAPHTEXT introduces minimal costs, there's potential for misuse in malicious recommendation systems and malware.

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

## REPRODUCIBILITY STATEMENT

The code to reproduce our results will be available soon. Our experimental settings and implementation details are stated in Section A.1, and the important hyper-parameters are discussed in Appendix A.3 section.

## A EXPERIMENTAL SETTINGS

### A.1 IMPLEMENTATION DETAILS

In our experiments node classification is approached as a multi-choice QA problem, and we employ the prompt detailed in Appendix B. The raw text attributes can be directly leveraged to form the textual information $F$. Additionally, we utilize the dataset metadata to extract raw text labels, creating a textual feature for each node. Nodes without data are assigned the value "NA". Consequently, every general graph can be perceived as a text-attributed graph with a minimum of one type of text attribute, namely the label. For continuous attributes, we use K-means to discretize continuous features (with the $K$ being the number of classes).

During the in-context-learning experiments, to prevent meaningless demonstrations lacking neighboring labels, we choose a sample with the highest degree from each label set. For the instruction tuning experiments using open-source LLMs, we can leverage continuous attributes in a more flexible way. Specifically, inspired by multi-modality LLMs Wang et al. (2023b), we use a Multi-Layer Perceptron (MLP) projector to map continuous features into the input text space, the token embedding space of LLaMA-2 (Touvron et al., 2023). We utilize AdamW (Loshchilov & Hutter, 2019) in conjunction with DeepSpeed (Rasley et al., 2020) to train the huggingface LLaMA2-7b model [2], with FP16 activated.

### A.2 DATASETS

Table 5: The statistics of the datasets.

| Benchmarks | #Nodes | #Edges | #Classes | #Features | #Train | #Validation | #Test |
|---|---|---|---|---|---|---|---|
| Cora | 2708 | 5278 | 7 | 1433 | 140 | 500 | 1000 |
| Cora-TAG | 2708 | 5278 | 7 | 1433 | 1624 | 541 | 543 |
| Citeseer | 3327 | 4552 | 6 | 3703 | 120 | 500 | 1000 |
| Citeseer-TAG | 3327 | 4552 | 6 | 3703 | 1911 | 637 | 638 |
| Cornell | 183 | 298 | 5 | 1703 | 87 | 59 | 37 |
| Texas | 183 | 325 | 5 | 1703 | 87 | 59 | 37 |
| Wisconsin | 251 | 515 | 5 | 1703 | 120 | 80 | 51 |

In this section, we provide more relevant details about the datasets we used in experiments. The dataset statistics are provided in Table 5. The datasets can be categorized into citation network datasets (i.e. Cora, Citeseer, and ogbn-arxiv) and web-page networks (i.e. Cornell, Texas, and Wisconsin), additionally, we use two text-attributed-graph (TAG) version of Cora and Citeseer, denoted as Cora-TAG and Citeseer-TAG.

**Citation graphs**: Most GNN-related studies, as referenced in works like Kipf & Welling (2017); Velickovic et al. (2018), often employ citation networks as benchmarking tools. Within these networks, nodes represent papers from the computer science domain. The features of these nodes are derived from bag-of-word vectors of the respective paper titles. Edges depict the citation links between these papers, while labels indicate the paper's specific categories. The text attributes are the title and abstract of the paper.

**WebKB graphs** Pei et al. (2020): Sourced by Carnegie Mellon University, aggregates web pages from computer science departments across several universities. We employ three specific subsets from this collection: Cornell, Texas, and Wisconsin. In these subsets, each node symbolizes a

---

[2]https://huggingface.co/meta-llama/Llama-2-7b-hf

web page, while edges denote hyperlinks connecting them. Nodes are characterized by a bag-of-words representation derived from their respective web pages. These pages have been meticulously categorized into five distinct classes: student, project, course, staff, and faculty.

The datasets mentioned above can be found in the following URLs: Cora [3], Citeseer [4], Cora-TAG [5], Citeseer-TAG [6], Texas [7], Cornell [8], Wisconsin [9].

### A.3 HYPERPARAMETERS

In GRAPHTEXT, the selection of text attributes $F$ and relations $R$ are the most important parameters. Here we discuss their choices and report the selected parameters in Table 6. For text attributes $F$, there are several choices: propagated features $A^k X$, propagated labels $A^k Y_L$, raw features $X$ and labels $Y_L$; For relationship $R$, there are several choices: k-hop shortest-path distance, denoted as $S_k$, propagated feature similarity, denoted as $\text{sim}(A^k X)$, and pagerank matrix Klicpera et al. (2019), with restart probability $\alpha = 0.25$, denoted as $\Pi$.

Table 6: GRAPHTEXT in-context learning hyperparameters.

|  | Text Attributes | Relations |
|---|---|---|
| **Cora** | $A^2 Y_L, A^3 Y_L$ | $S_0, \Pi, \text{sim}(A^2 X), \text{sim}(A^3 X)$ |
| **Citeseer** | $X, A^3 Y_L$ | $S_0, S_2, \Pi, \text{sim}(A^2 X)$ |
| **Texas** | $A^2 Y_L, A^3 Y_L$ | $S_2$ |
| **Wisconsin** | $X, Y_L$ | $\text{sim}(X), S_3$ |
| **Cornell** | $A Y_L, A^4 Y_L$ | $S_1, \text{sim}(A^3 X)$ |

## B PROMPT EXAMPLES

### B.1 FEW-SHOT IN-CONTEXT LEARNING

**Example of Citeseer:**
**[Human]:** You are a helpful assistant that classifies the topic of an academic paper based on the labels of the cited papers. You are going to choose the correct answer from several choices of paper categories: [A: Agents, B: Artificial Intelligence, C: Database, D: Information Retrieval, E: Machine Learning, F: Human Computer Interaction]

Here are a few examples:
<information>
    <third-order_pseudo_labels>
        <center_node>['A']</center_node>
        <1st_feature_similarity_graph>['A', 'A', 'A']</1st_feature_similarity_graph>
        <ppr>['A', 'B', 'A']</ppr>
    </third-order_pseudo_labels>
</information>
<question>What's the topic of academic paper given the information above?</question>
<answer>A</answer>

Remaining examples . . .

---

[3] https://relational.fit.cvut.cz/dataset/CORA
[4] https://linqs.soe.ucsc.edu/data
[5] https://github.com/CurryTang/Graph-LLM
[6] https://github.com/CurryTang/Graph-LLM
[7] https://docs.dgl.ai/en/0.8.x/api/python/dgl.data.html#node-prediction-datasets
[8] https://docs.dgl.ai/en/0.8.x/api/python/dgl.data.html#node-prediction-datasets
[9] https://docs.dgl.ai/en/0.8.x/api/python/dgl.data.html#node-prediction-datasets

Now let's answer the question below:
<information>
    <third-order_pseudo_labels>
        <1st_feature_similarity_graph>['C', 'B', 'B']</1st_feature_similarity_graph>
        <ppr>['C']</ppr>
    </third-order_pseudo_labels>
</information>

What's the topic of the paper given the information above? Valid choices are [A: Agents, B: Artificial Intelligence, C: Database, D: Information Retrieval, E: Machine Learning, F: Human computer interaction]. Remember, your answer should be in the form of the class choice wrapped by <answer></answer>.

[**Assistant**]: <answer>C</answer>

## B.2 INSTRUCTION TUNING

**Example of Cora:**
[**Human**]: Your goal is to perform node classification. You are given the information of each node in a xml format. Using the given information of a node, you need to classify the node to several choices: [<c0>: Rule_Learning, <c1>: Neural_Networks, <c2>: Case_Based, <c3>: Genetic_Algorithms, <c4>: Theory, <c5>: Reinforcement_Learning, <c6>: Probabilistic_Methods]. Remember, your answer should be in the form of the class label.
<information>
    <feature>
        <center_node><x><x emb></x></center_node>
        <1st_feature_similarity_graph><x><x emb></x></1st_feature_similarity_graph>
    </feature>
</information>

[**Assistant**]: The answer is: <c6>

Note that the "<x emb>" is the text token embedding for feature feature "x" generated by the MLP projector discussed in Appendix A.1.

## B.3 EXAMPLES OF TEXAS

**Node # 132, 136, 143, …**
Graph information:
pseudo labels:
center-node:['D']
second-hop neighbor:['D', 'D', 'D', 'D', 'D']
Target class: D

**Node # 30**
Graph information:
pseudo labels:
center-node:['D']
second-hop neighbor:['D', 'E', 'D', 'D', 'D']
Target class: D

**Node # 158**
Graph information:
pseudo labels:

> center-node:['A']
> second-hop neighbor:['A']
> Target class: A

We can observe that for the best setting in the Texas datasets, with hyperparameters discussed in Table 6, the center-node pseudo labels mostly assemble the second-hop neighbors. Consequently, removing the text information, i.e. removing the internal nodes in the graph-syntax tree in Section 4.3 does not hurt the performance.

This also shows the advantage of decoupling depth and scope Zeng et al. (2021) in the graph-syntax tree of GRAPHTEXT, which explains the performance gain of GRAPHTEXT over standard GNNs, e.g. GCN and GAT. A similar observation is also drawn in Figure 8 (i) of (Chien et al., 2021), where $A^2$ serves as the most important high-order aggregation scheme for Texas dataset.

## C  INTERACTIVE GRAPH REASONING

Since GRAPHTEXT facilitates graph learning within a textual domain, it allows for direct interaction between both humans and AI agents. In this section, we spotlight the interactive graph reasoning capabilities of GRAPHTEXT using a practical example. First, we demonstrate how GRAPHTEXT can engage in self-interaction via zero-shot chain of thought reasoning. Following that, we illustrate how human interactions can guide GRAPHTEXT to refine its graph reasoning approach.

### C.1  ZERO-SHOT CHAIN OF THOUGHT REASONING

Below is the example of graph reasoning on Cora node #2188 in the setting of standard zero-shot chain of thought reasoning Wei et al. (2022)[10] The input prompt for "Cora node #2188" is as below:

> **Input Prompt Cora node #2188:**
> **[Human]**: You are a helpful assistant that generates a classifies the topic of an academic paper based on the labels of the cited papers. You are going to choose the correct answer from several choices of paper categories:[A: Theory, B: Reinforcement Learning, C: Genetic Algorithm, D: Neural Network, E: Probabilistic Method, F: Case Based, G: Rule Learning]
> Here are a few examples:
> Graph information:
> pseudo labels:
> center-node:['F']
> ppr:['A', 'A', 'A', 'A']
> Topic of paper: A
>
> Graph information:
> pseudo labels:
> center-node:['B']
> ppr:['B', 'B', 'B', 'B']
> Topic of paper: B
>
> Graph information:
> pseudo labels:
> center-node:['C']
> ppr:['C', 'C', 'C', 'C']
> Topic of paper: C
>
> Graph information:

---

[10]ChatGPT and GPT-4 perform reasoning with explanation by default. Therefore, there is no necessity to add the additional "Let's think step-by-step" prompt for the text input.

```
pseudo labels:
center-node:['D']
ppr:['D', 'D', 'E', 'D']
Topic of paper: D

Graph information:
pseudo labels:
center-node:['A']
ppr:['E', 'E', 'A', 'A']
Topic of paper: E

Graph information:
pseudo labels:
center-node:['F']
ppr:['F', 'F', 'F', 'F']
Topic of paper: F

Graph information:
pseudo labels:
center-node:['G']
ppr:['G', 'G', 'G', 'G']
Topic of paper: G

Now let's answer the question below:
Graph information:
pseudo labels:
center-node:['G']
ppr:['A', 'G', 'A', 'E']
What's the topic of the paper given the information above? Valid choices are [A: Theory,
B: Reinforcement Learning, C: Genetic Algorithm, D: Neural Network, E: Probabilistic
Method, F: Case Based, G: Rule Learning]
```

As discussed in Section 4.2, according to the demonstrations, the PPR pseudo-labels provide a more robust mechanism for paper topic prediction. Utilizing either a count of PPR labels followed by a majority vote, or merely referencing the foremost PPR label, consistently results in the correct categorization in the given examples. Hence, based on these graph inductive biases derived from samples, we can reasonably figure out correct topic of the target paper should be A, which not only is the first entry, but also the predominant label in the PPR pseudo-label sequence. Now, let's use ChatGPT and GPT-4 to answer the problem. We performed 15 times of evaluation and here we show one sample of correct and incorrect responses below:

**Correct Response without Human Interaction**
**[ChatGPT]**: Among the PPR labels, 'A' appears twice, 'G' appears once, and 'E' appears once. 'A' is the most frequent label, so we classify the topic of the paper as 'A.'
Therefore, the topic of the paper is A: Theory.

**[GPT-4]**: Given the examples provided:
1. Whenever the center-node label is consistent with the majority of the PPR pseudo labels, the topic of the paper matches the center-node label.
2. However, when there's a discrepancy between the center-node label and the majority of the PPR labels, the topic of the paper is typically determined by the first (most crucial) label in the PPR list.
Given the graph information provided:
- The center node is labeled 'G' which corresponds to 'Rule Learning'.
- The 'ppr' nodes are labeled 'A', 'G', 'A', and 'E', which corresponds to 'Theory', 'Rule Learning', 'Theory', and 'Probabilistic Method' respectively.

> It seems that while the center node is 'G', the cited papers are mostly of category 'A' and 'E'.
> From the pattern observed in the examples, the category that has the majority or at least equal emphasis with other categories in the 'ppr' nodes tends to be the category of the paper. Therefore, in this case, 'A' or 'Theory' has the highest count in the 'ppr' nodes.
> Thus, the topic of the paper given the information above is: A: Theory.

> **Wrong Response without Human Interaction**
> **[ChatGPT]**: Based on the labels in the "ppr" list, we can see that 'A' (Theory), 'G' (Rule Learning), and 'E' (Probabilistic Method) are mentioned. However, the 'center-node' label is 'G' (Rule Learning), which indicates a stronger association with "Rule Learning." So, the topic of the paper is most likely "Rule Learning (G)."
>
> **[GPT-4]**: Given the graph information: pseudo labels: center-node:['G'] ppr:['A', 'G', 'A', 'E']
> The center node label is 'G', which corresponds to "Rule Learning". The papers referenced (ppr) are mixed between "Theory", "Rule Learning", and "Probabilistic Method". However, since the center node holds the primary topic label, the academic paper's topic is: G: Rule Learning.

From the analysis above, it's clear that:

By recasting the graph learning challenge into problems that both humans and LLMs can interpret and reason with, GRAPHTEXT facilitates LLMs in graph reasoning. For instance, ChatGPT can deduce the right answer through label counting, as highlighted in green. Impressively, GPT-4 discerns the correct solution by employing the Personalized PageRank (PPR) prior, as illustrated in blue, and identifies the first PPR neighbor as the most important factor for prediction.

However, the lucid explanations rendered in natural language allowed us to pinpoint certain inadequacies in their graph reasoning approach:

**Failure to Retrieve Prior Knowledge**: Table 4.1 shows that LLMs, particularly ChatGPT, don't always effectively recall their previous understanding of graphs; only 26.7% of ChatGPT's reasoning relies on PPR.

**Persistence in Prior Belief**: Despite the inherent reasoning prowess of LLMs, they can sometimes be hindered by preconceived notions about graphs. A detailed examination of the results in Table 4.1, accompanied by the explanations of reasoning above, indicates that a deep-seated bias - where the central node is viewed as the most vital (denoted in red) - is a primary cause of mistakes.

In the following section, we engage with GRAPHTEXT to enhance its graph reasoning capabilities.

## C.2    GRAPH REASONING WITH HUMAN INTERACTION

As highlighted in the preceding section, LLMs occasionally struggle with recalling prior knowledge and remain anchored to their pre-existing beliefs. To address this, we engage with GRAPHTEXT by: (1) providing a clear definition of PPR, and (2) explicitly instructing them to assess the reliability of both center-node pseudo and PPR pseudo labels. This leads to the following message:

> **Human Interaction** Note that PPR pseudo labels represent a sequence of labels ranked by importance, with the first one being the most crucial. Let's evaluate the reliability of center node pseudo labels against PPR pseudo labels and then make another prediction.

With the interaction above, in most cases LLMs refined their answer, below are some examples:

> **Correct Reasoning after Human Interaction**
> **[ChatGPT]**: In this PPR list, the labels 'A' and 'G' are present, with 'A' appearing before

'G,' which means 'A' is considered more crucial in the context of this paper. Now, let's make a prediction based on this evaluation: Since 'A' is the most crucial label in the PPR list, we will prioritize it for classifying the topic of the paper. Therefore, the paper's primary topic is "A: Theory" (Category A). The presence of 'G' in the PPR list indicates some relevance to "Rule Learning" (Category G), but because it comes after 'A,' it is considered less important in this context. So, the revised prediction is that the primary topic of the paper is "A: Theory."

**[GPT-4]**: Given the new information that PPR pseudo labels represent a sequence of labels ranked by importance, let's re-evaluate the provided data:
The center node suggests the paper is about 'G' (Rule Learning). However, considering the prominence of 'A' (Theory) in the ranked sequence, and since it's the most crucial label, the paper might be primarily focused on theoretical aspects related to Rule Learning. Considering both the center node and the importance sequence, the topic of the paper should be: A: Theory

**Refusion after Human Interaction**

**[ChatGPT]**: Certainly, let's evaluate the reliability of center node pseudo labels against PPR pseudo labels.
In the provided graph information: The center-node label is 'G,' which suggests that the most important label for this paper is 'G.' However, when we consider the PPR labels, we see that 'A' appears as the first and third label, which might indicate some importance as well.
Let's consider both the center-node and the PPR labels for our prediction. Given that 'G' is the center-node label and 'A' appears in the PPR labels, we can prioritize the center-node label but also consider the presence of 'A' in the PPR labels.
So, the prediction for the topic of the paper can be: Topic of paper: G (with some influence from A).

**Confusion after Human Interaction**

**[ChatGPT]**: Based on this information, we can classify the topic of the paper as a combination of the most frequent labels among the neighboring papers, which are 'A' and 'G'. Therefore, the topic of the paper is likely a combination of topics 'A' and 'G'.
So, the answer is a combination of two topics, and the closest match among the given options is:
Option: F: Case Based *# Note that F does not present in the pseudo labels of the question*

The provided examples, along with the consolidated findings in Table 4.1, compellingly show that both ChatGPT and GPT-4 can adjust their pre-existing biases about graphs when given human feedback. Notably, after this interaction, GPT-4 delivers a flawless accuracy rate of 100%, consistently following the PPR logic. Meanwhile, ChatGPT also sees a significant performance boost, with an accuracy improvement of 36.9%. However, as evidenced in the examples, ChatGPT occasionally refuses updating its predictions or becomes confused.

