# OpenReview forum: "GraphText: Graph Learning in Text Space"
_ICLR.cc/2024/Conference — ICLR 2024 Conference Withdrawn Submission_

### Official Review · Reviewer_JsWa · 2023-10-15

**Soundness:** 3 good
**Presentation:** 2 fair
**Contribution:** 3 good
**Rating:** 5
**Confidence:** 5

**Summary:**

This paper introduces an innovative method for translating graph-structured data into a natural language that Large Language Models  (LLMs) can understand. The authors demonstrate that their proposed approach facilitates training-free graph reasoning and enables interactive graph-based reasoning.

**Strengths:**

- **Innovative Graph-Syntax Tree Design**: One of the notable strengths of this paper is the introduction of the graph-syntax tree. Particularly impressive is the discretization of continuous node features. This novel approach is significant as it strikes a balance between providing informative features to LLMs while avoiding the issue of massive input tokens. Besides, this creative design facilitates feature propagation and thus significantly enhances LLMs' graph reasoning capabilities.

- **Comprehensive Experimental Evaluation:** The authors demonstrated the effectiveness and flexibility of the proposed method by testing it in different scenarios. These include training-free settings with closed-source LLMs, fine-tuning with open-source LLMs, interactive graph reasoning, etc. This comprehensive evaluation underscores the practical utility and versatility of the proposed approach, making it a valuable contribution to the field.

**Weaknesses:**

- **Heavy Parameter Tuning**: The paper's heavy reliance on dataset-specific parameter tuning, as demonstrated in Table 6, raises concerns about the generalizability and effectiveness of the proposed method. Conducting ablation studies on the selection of these hyperparameters would be beneficial to determine whether the performance boost is primarily a result of heavy parameter search or the inherent design of the method itself. This clarification would help assess the true impact of the proposed method.


- **Reliance on Domain Knowledge for Interactive Graph Reasoning**: While the paper successfully highlights the adaptability of LLMs to human feedback for interactive graph reasoning, it is important to recognize that this approach heavily depends on the quality and relevance of the human feedback. Not all tasks can be addressed using a single inductive bias, such as the PPR label. This necessitates tailoring human feedback for specific graph tasks.

**Questions:**

- Q1: Could you provide information on the number of prompts/training samples used in the experiment presented in Table 1?

- Q2: When transforming the continuous features into a discrete space using K-means, it would be valuable to know the specific value of K that was used in your experiments. Additionally, could you explain the heuristic or rationale behind selecting this particular value for K?

---

> ### Author Response · Authors · 2023-11-17
> **Response to Reviewer JsWa**
>
> We sincerely appreciate your insightful comments. Below, we address each of your points:
>
> ### W1.Heavy Parameter Tuning
> We would kindly argue that in fact, GraphText has lighter parameter tuning compared with standard GNN framework. It has many advantages due to the training-free property of the framework for ICL:
> 1. **Smaller search space**: For a basic machine learning model, besides the model parameters, we typically have to determine the hyperparameters for training, including the number of epochs, learning rate, batch size, learning rate schedule, weight decay, and regularization term. Since there is no graph training, these hyperparameters related to model are not searched. The **only two hyperparameters** for GraphText to search are the choices of text information and relation information to construct graph-syntax trees. As the size of the hyperparameter search space grows exponentially in terms of the number of hyperparameters, GraphText enjoys a much smaller search space.
> 2. **No need for training multiple models for multiple datasets**: Only one model, i.e. ChatGPT is used, compared with 5 individual models trained for each dataset.
> 3. **Less device requirement**: Since there is no training, and we can leverage closed-end API calling of ChatGPT, no GPU is required to perform GraphText-ICL experiments.
> 4. **Cheap inference**: The cost is actually extremely low, for example, for each sample, empirically, it takes around a total of 2.5 hour (with 4 parallel workers) for the hyperparameter search on three heterophilic datasets (15 USD for 300 runs).
>
> ### W2: Reliance on Domain Knowledge for Interactive Graph Reasoning
>
> We acknowledge the importance of domain knowledge in interactive graph reasoning. However, it's worth noting that human feedback is an optional enhancement rather than a necessity for our model. Except for the experiments in Section 4.2, all other tests were conducted in a single-turn question-answering format without any human feedback. This setup demonstrates the robustness and versatility of GraphText in handling various graph-related tasks.
> ### Q1: Number of Prompts/Training Samples in Experiments
> In the ICL experiments presented in Table 1, we selected one example per class from the training set, ensuring a fair selection by choosing the example with the largest degree to avoid biased selection (e.g., orphan nodes). Please refer to Appendix C.1 for a concrete example.
> ### Q2: K-means Parameter and Rationale
> For transforming continuous features into a discrete space using K-means, we set K equal to the number of classes. This choice is based on the intuition that if raw or propagated features can be clustered into C distinct clusters representing each class, the LLM can easily learn a direct one-to-one match through in-context learning.
>
> We hope that our detailed responses have satisfactorily addressed the concerns and questions raised by the reviewers.

---

> > ### Comment · Reviewer_JsWa · 2023-11-22
> >
> > Thank you for your response and addressing my questions. I agree with the answer to W2.
> >
> > Regarding W1, the statement "The only two hyperparameters for GraphText to search are the choices of text information and relation information" is not convincing to me. For Text Attributes, there are mutiple choices, such as whether to include X, the value of k for A^k, different conbimation of A^k. Similarly, for Relations, there are multiple choices for the value of k for S_k, A^k, and decisions on which/how many of them should be included. Furthermore, the claim of cheap inference is based on using small datasets with fewer than 4,000 nodes. However, when dealing with larger datasets like ogbn-arxiv, such parameter searches become impractical.
> >
> > I incline to maintain my score.

---

### Official Review · Reviewer_LfFz · 2023-10-19

**Soundness:** 3 good
**Presentation:** 3 good
**Contribution:** 3 good
**Rating:** 8
**Confidence:** 4

**Summary:**

In this paper, the authors introduce GRAPHTEXT, a novel framework designed to bridge the gap between Large Language Models (LLMs) and graph machine learning. While LLMs have excelled in natural language understanding and reasoning, they have faced challenges in applying their capabilities to graph-structured data. GRAPHTEXT addresses this issue by translating graphs into natural language, enabling LLMs to perform graph reasoning tasks. GRAPHTEXT presents a promising approach to extend the capabilities of LLMs into the realm of graph machine learning, offering potential benefits for various applications that involve graph-structured data.

**Strengths:**

1. The paper presents an elegant solution to a fundamental problem: How to derive a language for relational data. The proposed tree-based solution provides a principled way to bridge relational data and one-dimensional sequential language. This innovation has the potential to catalyze significant future research.

2. GraphText equips LLMs with the ability to reason over graphs using natural language, thereby enabling interactive graph reasoning. The distinct aspects of interpretability and interactiveness of GraphText differentiate it from traditional GNNs.

3. Another outstanding feature of GraphText is its training-free reasoning ability, which not only reduces the computational overhead but also delivers impressive performance, even surpassing some supervised GNNs. Such capabilities indicate great potential for real-world applications.

**Weaknesses:**

1. A comparative analysis with existing baselines is required. It would be especially beneficial to compare GraphText against other methods like GraphML and GML, which explore a similar problem

2. How to construct discrete text from continuous features is not comprehensively studied. According to the hyper-parameters, the best settings are mostly based on label propagation.

3. The algorithm is overall good. Nonetheless, there is a lack of time complexity analysis. I think it should be added then.

**Questions:**

See in weakness

---

> ### Author Response · Authors · 2023-11-17
> **Response to Reviewer LfFz**
>
> We appreciate your insightful comments and enhanced our manuscript based on your feedback. Below, we address each of your points:
> ### W1: Comparative Analysis with Existing Baselines
> We understand the importance of benchmarking our method against established baselines in the field. Accordingly, we have added additional In-Context Learning (ICL) baselines, including GraphML and GML, in our updated results presented in Table 1. The comparative analysis demonstrates that GraphText not only holds its ground against these methods but also achieves state-of-the-art (SOTA) ICL performance. We report an absolute performance gain of 34.02% over the best ICL baselines, which significantly underscores the effectiveness and innovation of GraphText in the context of graph-based learning.
> ### W2: Construction of Discrete Text from Continuous Features
> The core objective of our research is to establish a bridge between the text and graph domains using a tree-based prompt framework. The task of discretizing continuous features into textual format, while vital, is a complex endeavor that we have identified as a potential avenue for future research. We acknowledge that the optimal settings in our current model are predominantly reliant on label propagation, and we aim to explore more advanced discretization techniques as future work.
> ### W3: Time Complexity Analysis
> The time complexity for obtaining a graph's label distribution reveals distinct operational differences is shown in the table below:
>
> |           | Preprocessing | Training         | Inference      |
> | --------- | ------------- | ---------------- | -------------- |
> | GCN       | N/A           | $O(PEF + PNF^2)$ | $O(EF + NF^2)$ |
> | GraphText | $O(N^2F+NE)$  | N/A              | $O(NT^2)$      |
>
>
> Specifically, a typical GNN, undergoes three stages: (1) Preprocessing the graph, (2) Training on the graph, and (3) Inference. For a graph with $N$ nodes and $E$ edges, where label, features, and hidden dimensions are uniformly dimension $F$, the time complexity for a $L$-layer GCN message passing on $F$ is $O(LEF + LNF^2)$, simplifying to $O(EF+NF^2)$. This complexity applies to both training and inference. However, training usually requires more wall time due to the need for repeated forward passes and gradient updates for multiple epochs until convergence.
>
> In contrast, the GraphText framework, being training-free, eliminates training time entirely. It operates in two phases: a preprocessing phase, where synthetic textual and relational data are generated to construct graph-syntax trees, and an inference phase, where LLM inference is carried out.
>
> The time complexity of preprocessing depends on the chosen synthetic text and relation information for the graph-syntax tree. For synthetic text, feature propagation in the synthetic text information phase takes $O(LEF)$ time. The $K$-means algorithm, used here, has a time complexity of $O(NKFI)$ ($I$ being the number of iterations).
>
> For synthetic relations, calculating row-wise similarity of the feature matrix requires $O(N^2F)$ time. Finding the Shortest Path Distance (SPD) with a maximum hop $H$ (empirically set to $H \leq 4$), optimally solved using Bellman Ford [1], results in $O(EH)$ time complexity for one node and $O(NEH)$ for all nodes.
>
> Thus, the total time complexity for preprocessing, combining all synthetic relations, is $O(LEF+NKFI+N^2F+NEH)$, simplifying to $O(N^2F+NE)$. This is manageable since preprocessing is a one-time process.
>
> The inference time depends solely on the LLM's inference time, influenced by the context length $T$, model size, inference speedup techniques, and API response time (if using an API like OpenAI models). For instance, using ChatGPT for 600 tokens approximately takes 1 second.
>
> In conclusion, GraphText, unlike GNNs, has no training time, with its primary time complexity bottleneck in preprocessing. Users can adjust and select less computationally intensive synthetic text/relations for graph-syntax trees, enabling GraphText's application on large graphs.

---

> > ### Comment · Reviewer_LfFz · 2023-11-20
> > **Thanks for your response**
> >
> > Thanks for your response. I have no more concerns and raise my score accordingly.

---

### Official Review · Reviewer_ekj8 · 2023-10-30

**Soundness:** 3 good
**Presentation:** 3 good
**Contribution:** 2 fair
**Rating:** 3
**Confidence:** 4

**Summary:**

The paper addresses an important problem of bridging the gap between LLM and GNN. A GRAPHTEXT method is introduced to construct a graph-syntax tree by incorporating node features and structural information in a graph. The graph-syntax tree is then converted to a graph text sequence, which is then processed by an LLM to treat graph-related tasks as text generation tasks. However, the experiments seem insufficient in some aspects.

**Strengths:**

Paper Strength:

1. Exploring the gap between LLM and GNN is highly meaningful, as it effectively utilizes the potential of LLMs.

2. The proposed graph-syntax tree is novel and conceptually sound.

**Weaknesses:**

Paper Weakness:

1. The article lacks a comparison with large text-attributed graph datasets such as OGB-Arxiv, which is a commonly-used dataset extensively discussed in numerous papers regarding text-attributed graphs [1-4].

2. Regarding the experiments, I have several questions. (1) Since Wisconsin, Texas, and Cornell datasets are relatively small datasets, it would be beneficial if the authors could provide variance analysis of the results. The results on these datasets may exhibit considerable variance. (2) Moreover, Wisconsin, Texas, and Cornell datasets exhibit high heterophily. Therefore, when evaluating performance on these datasets, it is crucial to include comparisons with MLP. Based on the results from prior studies [5], it appears that GraphText may not show a significant advantage over MLP. (3) There appears to be an inconsistency in the accuracy of GraphText on Cora between Table 3 and Table 1. Could you provide an explanation for this inconsistency? (4) Providing the results under different hyper-parameters regarding the selection of text attributes and relations in Appendix A.3 would be valuable. (5) The comparison with directly utilizing LLMs to handle text attributes while ignoring the graph structure is essential.

3. It would be helpful if the authors can provide a time complexity analysis for constructing the graph-syntax tree and run-time requirements. It seems that it would be costly to construct a graph-syntax tree on large-scale graphs.

4. Can the proposed framework help other graph-related tasks, like link prediction, community detection?

[1] Chen, Zhikai, et al. Exploring the potential of large language models (llms) in learning on graphs.

[2] Duan, Keyu, et al. Simteg: A frustratingly simple approach improves textual graph learning.

[3] Zhao, Jianan, et al. Learning on large-scale text-attributed graphs via variational inference.

[4] He et al. Harnessing Explanations: LLM-to-LM Interpreter for Enhanced Text-Attributed Graph Representation Learning.

[5] Zhu, Jiong, et al. Beyond homophily in graph neural networks: Current limitations and effective designs.

**Questions:**

See weakness.

---

> ### Author Response · Authors · 2023-11-17
> **Response to Reviewer ekj8 [1/2]**
>
> We appreciate your valuable feedback and have addressed each of your points as follows:
> ### W1: Comparison with Large Text-Attributed Graph Datasets
> General graphs with continuous features are much more common and challenging for LLM to perform reasoning on compared with text-attributed graphs (TAGs), which demand a text description for each node. For example, for Cora, the problem of classification of a paper’s topic becomes much easier once the node's text attributes (title and abstract) are given compared with using only the continuous feature and observed labels. Hence, we believe that **general graphs are better datasets for evaluating LLM’s graph reasoning ability than TAGs**. In addition, we also have TAG experiments on Section 4.4, showing that GraphText is effective in TAGs.
>
> Regarding large-scale datasets, the experimental setting of in-context learning (ICL) distincts itself from standard supervised finetuned (SFT) GNNs. Due to the limitation of OpenAI API calling, hundreds/thousands of samples are evaluated in standard ICL literature [3,4,5] and are enough to prove the concept. The recent endeavor [1] also leverages 100 samples in their experiments. In our main experiments, we are presenting the full test set results of Cora and Citeseer (1000 nodes) along with three heterophilic datasets.
>
> ### W2: Questions
> - **Variance of Heterophilic Datasets**: The mean and variance of results (5 different seeds) are: Texas: (75.7, 0.0) Wisconsin(67.9, 0.9) Cornell (57.8, 1.5), which is similar to GCN’s: Texas (59.5, 4.1), Wisconsin (49.0, 0.6), Cornell (37.8, 1.2).
> - **Comparison with Supervised Baselines, e.g. MLP**: While we acknowledge that In-Context Learning (ICL) approaches typically do not outperform Supervised Fine-Tuning (SFT) methods, the primary objective of GraphText is to enhance ICL capabilities for Large Language Models (LLMs) in graph reasoning tasks. Our focus was not to surpass all existing SFT GNN baselines but to demonstrate the effectiveness of GraphText in an ICL setting. Notably, GraphText achieves state-of-the-art performance in this domain, realizing a significant absolute performance gain of 34.0% over the best existing ICL baselines. This advancement is noteworthy, especially considering that in some cases, GraphText even manages to outperform certain SFT GNN methods, highlighting its potential in ICL for graph-related tasks.
> - **Inconsistency in Accuracy on Cora**: We apologize for the discrepancy in the accuracy figures for Cora in Tables 3 and 1. The correct performance for Cora should be 68.3%. We have corrected it in the updated version.
> - **Hyper-parameter Study**: While Table 1 and 3  includes several ablation studies, a comprehensive hyper-parameter study was beyond our scope due to time constraints and will be considered for future research.
> - **Utilizing LLMs with General Graphs**: As our focus is on general graphs without text attributes, the proposed experiment comparing the direct use of LLMs on text attributes while ignoring graph structure is not feasible in our study.
> ### W3: Time Complexity Analysis
> Please see the other response for details.
>
> ### W4: Applicability to Other Graph Tasks
>
> Yes, GraphText can be applied to a range of graph-related tasks. By constructing graph-syntax trees tailored to specific tasks, the LLM can effectively address various scenarios. For instance, in link prediction tasks like recommendation systems, a graph-syntax tree can be formed with users and candidate items as nodes, enabling the LLM to predict user preferences.
>
>
> ### References
>
> [1] Chen, Zhikai, et al. "Exploring the potential of large language models (llms) in learning on graphs." arXiv preprint arXiv:2307.03393 (2023).
>
> [2] Polak, Adam. "Bellman-Ford is optimal for shortest hop-bounded paths." arXiv preprint arXiv:2211.07325 (2022).
>
> [3] Wei, Jason, et al. "Chain-of-thought prompting elicits reasoning in large language models." Advances in Neural Information Processing Systems 35 (2022): 24824-24837.
>
> [4] Wang, Xuezhi, et al. "Self-consistency improves chain of thought reasoning in language models." arXiv preprint arXiv:2203.11171 (2022).
>
> [5] Yao, Shunyu, et al. "Tree of thoughts: Deliberate problem solving with large language models." arXiv preprint arXiv:2305.10601 (2023).

---

> > ### Author Response · Authors · 2023-11-17
> > **Response to Reviewer ekj8 [2/2]**
> >
> > ### W3 Time Complexity
> > The time complexity for obtaining a graph's label distribution reveals distinct operational differences is shown in the table below:
> >
> > |           | Preprocessing | Training         | Inference      |
> > | --------- | ------------- | ---------------- | -------------- |
> > | GCN       | N/A           | $O(PEF + PNF^2)$ | $O(EF + NF^2)$ |
> > | GraphText | $O(N^2F+NE)$  | N/A              | $O(NT^2)$      |
> >
> >
> > Specifically, a typical GNN, undergoes three stages: (1) Preprocessing the graph, (2) Training on the graph, and (3) Inference. For a graph with $N$ nodes and $E$ edges, where label, features, and hidden dimensions are uniformly dimension $F$, the time complexity for a $L$-layer GCN message passing on $F$ is $O(LEF + LNF^2)$, simplifying to $O(EF+NF^2)$. This complexity applies to both training and inference. However, training usually requires more wall time due to the need for repeated forward passes and gradient updates for multiple epochs until convergence.
> >
> > In contrast, the GraphText framework, being training-free, eliminates training time entirely. It operates in two phases: a preprocessing phase, where synthetic textual and relational data are generated to construct graph-syntax trees, and an inference phase, where LLM inference is carried out.
> >
> > The time complexity of preprocessing depends on the chosen synthetic text and relation information for the graph-syntax tree. For synthetic text, feature propagation in the synthetic text information phase takes $O(LEF)$ time. The $K$-means algorithm, used here, has a time complexity of $O(NKFI)$ ($I$ being the number of iterations).
> >
> > For synthetic relations, calculating row-wise similarity of the feature matrix requires $O(N^2F)$ time. Finding the Shortest Path Distance (SPD) with a maximum hop $H$ (empirically set to $H \leq 4$), optimally solved using Bellman Ford [1], results in $O(EH)$ time complexity for one node and $O(NEH)$ for all nodes.
> >
> > Thus, the total time complexity for preprocessing, combining all synthetic relations, is $O(LEF+NKFI+N^2F+NEH)$, simplifying to $O(N^2F+NE)$. This is manageable since preprocessing is a one-time process.
> >
> > The inference time depends solely on the LLM's inference time, influenced by the context length $L$, model size, inference speedup techniques, and API response time (if using an API like OpenAI models). For instance, using ChatGPT for 600 tokens approximately takes 1 second.
> >
> > In conclusion, GraphText, unlike GNNs, has no training time, with its primary time complexity bottleneck in preprocessing. Users can adjust and select less computationally intensive synthetic text/relations for graph-syntax trees, enabling GraphText's application on large graphs.

---

> > > ### Comment · Reviewer_ekj8 · 2023-11-23
> > > **Thanks for your response**
> > >
> > > Thank you for your response. However, I still have some concerns. (1) The lack of performance evaluation on large-scale datasets, particularly the OGB-Arxiv dataset, remains a significant concern. The authors mentioned that the proposed method deals with continuous features, which limits the evaluation on OGB-Arxiv. I think a potential solution could be transforming text description into continuous features and subsequently adapting it within the proposed framework. Besides, regarding the limitation of OpenAI API calling, utilizing an open-source LLM could serve as a feasible alternative. (2) When comparing the complexities of GCN and GraphText, it becomes evident that the complexity of GCN remains relatively lower, even when GraphText does not involve a training phase. I maintain my initial score

---

### Official Review · Reviewer_2ufP · 2023-11-01

**Soundness:** 2 fair
**Presentation:** 2 fair
**Contribution:** 2 fair
**Rating:** 3
**Confidence:** 4

**Summary:**

This paper proposes a method GRAPHTEXT to solve graph tasks by LLMs. The method constructs a syntax tree to describe necessary information about the graph. Then the syntax tree is traversed to the prompt. LLMs can use the prompt to get information about the graph. The paper also proposes to use discretization methods like clustering to transform continuous feature into discrete space. Experiment results show that the method can achieve good performance in some datasets.

**Strengths:**

1. Bridging the gap between graphs and LLMs is an interesting and important problem.

**Weaknesses:**

1. The performance on Cora/CiteSeer is very low, According to [1], GPT3.5 can achieve 67% accuracy on Cora with target text features, but ChatGPT performance shown in Table is much worse than it (label+feat, original). I guess it is because of the prompt used. I suggest to use a better prompt for the baseline. Besides, comparing 67% with the highest 68.3% performance in Cora, the proposed method does not provide good benefit to the task.
2. Some datasets used (Texas, Wisconsin, Cornell) are heterophily graphs where many GNNs cannot outperform MLP. You should compare with MLP and heterophily graph methods as baselines.
3. Some essential parts are missing in the paper. See questions.
4. in Section 4.2, the observation 2 is confusing. Why in-context learning being good indicate that GPT-4 outperforms ChatGPT?

[1] Chen, Z., Mao, H., Li, H., Jin, W., Wen, H., Wei, X., ... & Tang, J. (2023). Exploring the potential of large language models (llms) in learning on graphs. arXiv preprint arXiv:2307.03393.

**Questions:**

1. How to generate the pseudo labels in Figure 2a?
1. Section 4.2 uses human feedback to improve LLM prediction, can you provide the details about human feedback?

---

> ### Author Response · Authors · 2023-11-17
> **Response to Reviewer 2ufP**
>
> We appreciate your insightful comments and concerns. Below, we address each of the weaknesses you highlighted:
>
> **W1: Performance on Cora/CiteSeer**
>
> We acknowledge the good performance of Cora and Citeseer reported in [1]. However, it is essential to clarify that [1] leverages text-attributed graphs, which inherently simplify LLM reasoning due to the presence of raw text attributes. For citation graphs like Cora and Citeseer, classifying a paper’s topic becomes a much easier problem once the node's text attributes (title and abstract) are given compared with using only the continuous feature and observed labels.  Our approach, on the other hand, targets a more complex scenario—general graphs (where only continuous features, i.e. bag of words in Cora and Citeseer, are available). This distinction is crucial as it places our method in a more challenging context compared to the text-attributed graph setting. To further ease the concerns, we include [1], denoted as NeighborText, as a baseline in the updated paper and leverage the observed label as raw text attributes. We can see that GraphText shows an average absolute improvement of 50.26%, which clearly shows the effectiveness of our design.
>
> **W2: Comparison with MLP and Heterophily Graph Methods**
>
> Your suggestion that MLP could achieve better heterophily results is true. However, it is important to consider the unique ICL setting of GraphText, where a single LLM for text generation addresses graph problems via reasoning from only one sample per class without any specific graph training. This ICL framework significantly differs from the supervised learning settings employed by GNN/MLP, where multiple nodes (up to 20 per class) are used for training, and different models are trained for each graph. Hence, it is unfair to compare SFT baselines with ICL baselines. Nevertheless, GraphText not only outperforms other ICL baselines significantly (by 34.02%) but also shows competitive results against several GNN baselines, achieving a 6.64% absolute performance gain over GCN, even without graph-specific training.
>
> **W3: Clarification on Pseudo Labels and Human Interaction**
>
> We apologize for the confusion, the pseudo labels are generated from $k$-hop propagated labels, i.e. $\boldsymbol{A}^{k}\boldsymbol{Y}_L$, (as discussed in Appendix A.3). The ‘human feedback’ states the definition of PPR, as illustrated in Figure 2, and let LLM re-evaluate the problem. To provide further clarity, we have included a detailed analysis of interactive graph reasoning in Appendix C. This analysis includes detailed prompts, human feedback, and sample reasoning from ChatGPT and GPT-4.
>
> **W4: In-Context Learning Performance of GPT-4 vs. ChatGPT**
>
> The short answer is: as shown in Table 2, for the same prompt of a graph reasoning problem, GPT-4 (73.3 accuracy) outperforms ChatGPT (26.7 accuracy) which indicates GPT-4 outperforms ChatGPT. For an in-depth discussion on this topic and how GPT-4 outperforms ChatGPT with better reasoning, we refer you to the expanded discussion in Appendix C.
>
> [1] Chen, Zhikai, et al. "Exploring the potential of large language models (llms) in learning on graphs." arXiv preprint arXiv:2307.03393 (2023).

---

### Author Response · Authors · 2023-11-17
**General Response to All Reviewers**

We gratefully acknowledge the reviewers for their constructive feedback and insightful suggestions. We have made several updates to our manuscript to address the common concerns and misunderstandings that were highlighted. Below, we provide detailed responses to the key issues raised:
1. In-Context Learning (ICL) vs. Supervised Fine-Tuning (SFT):
Since ICL approaches evaluate the generalization performance of a pre-trained model, it is a known fact that ICL typically underperforms compared to SFT methods specifically tuned on the dataset. Therefore, the primary objective of our study was not to surpass all existing SFT GNN baselines but to demonstrate the effectiveness of our proposed GraphText approach within the ICL framework. Our results show that GraphText enables Large Language Models (LLMs) to perform graph reasoning in text space without training on graphs, achieving state-of-the-art ICL performance with an average improvement of 34.0% over the best ICL baselines, even surpassing some SFT GNN methods sometimes. This significant advancement aligns with our goal to enhance ICL capabilities.
2. Text-Attributed Graphs (TAGs) vs. General Graphs:
Please kindly note that the frequently mentioned work [1] are designed for TAGs, which include a text description for each node, and are more amenable to LLM applications compared to general graphs with continuous features. For example, for Cora, the problem of classification of a paper’s topic becomes much easier once the node's text attributes (title and abstract) are given compared with using only the continuous feature and observed labels. Meanwhile, our method is designed to tackle the more challenging setting of general graphs. Hence, it's crucial to note that comparing TAG results in [1] to our results from general graphs would be unfair.
To further ease the concerns, we compare [1] with GraphText on general graphs (where the raw text is the observed labels). GraphText shows an average absolute improvement of 50.26%, which clearly shows the effectiveness of our design.

In response to the reviewers' comments, we have updated our manuscript:
- Section 4.1 Enhancements: We have included GraphML, GML [2], and NeighborText [1] as additional ICL baselines. This inclusion provides a more comprehensive comparison and underscores the advancements made by our GraphText approach.
- Expanded Discussion in Appendix C: We have added a detailed explanation and discussion on how human interaction can enhance LLM reasoning. This expansion addresses the queries raised about the role of human interaction in our framework.
We hope that these updates and clarifications address the concerns of the reviewers. We are committed to advancing the field and believe that our work on GraphText marks a significant step in this direction. We sincerely thank the reviewers for their invaluable feedback, which has helped us improve the quality and clarity of our research.


[1] Chen, Zhikai, et al. "Exploring the potential of large language models (llms) in learning on graphs." arXiv preprint arXiv:2307.03393 (2023).

[2] Guo, Jiayan, Lun Du, and Hengyu Liu. "GPT4Graph: Can Large Language Models Understand Graph Structured Data? An Empirical Evaluation and Benchmarking." arXiv preprint arXiv:2305.15066 (2023).

---

### Author Response · Authors · 2023-11-21
**Response to All Reviewers**

Dear Reviewers,

Thank you once again for your invaluable effort and time in reviewing our paper. As we approach the deadline for revisions and discussions, we eagerly anticipate any further feedback you may have.

We believe that the primal proposed weaknesses of reviews are based on the inappropriate comparisons of our ICL results against supervised learning results, especially against supervised text-attributed graph results. We hope our detailed clarification would ease these concerns.

Should you have any remaining questions or find any aspect of our responses unclear, please do not hesitate to reach out. We are fully prepared to engage in further discussion regarding any technical details or concerns you may have. Conversely, if our revisions have met your expectations, we kindly request that you reconsider your evaluation of our work.

We greatly appreciate your contributions to improving our research and look forward to your thoughts.